# In Vivo Fibroblast Activation of Systemic Sarcoidosis: A ^68^Ga-FAPI-04 PET/CT Imaging Study

**DOI:** 10.3390/diagnostics13081450

**Published:** 2023-04-17

**Authors:** Jingnan Wang, Li Huo, Lu Lin, Na Niu, Xiang Li

**Affiliations:** 1Department of Nuclear Medicine, Peking Union Medical College Hospital, Chinese Academy of Medical Sciences, Beijing 100730, China; 2Beijing Key Laboratory of Molecular Targeted Diagnosis and Therapy in Nuclear Medicine, Beijing 100730, China; 3Department of Radiology, Peking Union Medical College Hospital, Chinese Academy of Medical Sciences, Beijing 100730, China; 4Division of Nuclear Medicine, Department of Biomedical Imaging and Image-Guided Therapy, Medical University of Vienna, 1010 Vienna, Austria

**Keywords:** ^68^Ga-FAPI-04 PET, cardiac MR, sarcoidosis

## Abstract

A 47-year-old female with cardiac dysfunction and lymphadenopathy underwent ^18^FDG PET/CT and ^68^Ga-FAPI-04 imaging for tumor screening. Mild uptake in the left ventricular wall was detected on the oncology ^18^FDG PET/CT. True myocardiac-involvement could not be distinguished with physiological uptake. The following ^68^Ga-FAPI-04 showed intense heterogeneous uptake in the left ventricular wall, particularly in the septum and apex area, corresponding with the late gadolinium enhancement regions shown by cardiac MR. Intense uptake was also noted in the mediastinal and bilateral hilar lymph nodes. Endomyocardial biopsy demonstrated sarcoidosis.

**Figure 1 diagnostics-13-01450-f001:**
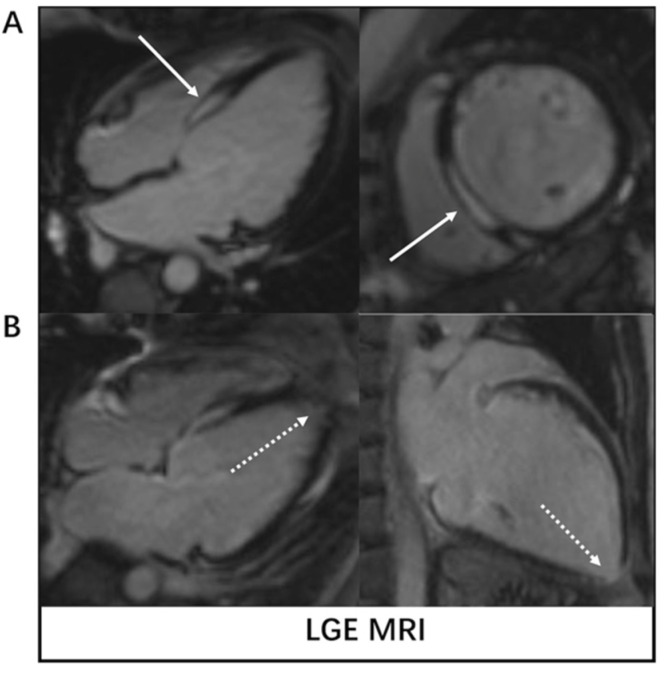
A 47-year-old female presented with intermittent fever, shortness of breath, and aggravated edema of the lower limbs over 4 months. Elevated N-terminal prohormone of brain natriuretic peptide (NT-proBNP) of 2450 pg/mL was recorded. The cardiac MR (CMR) images showed left ventricular (LV) enlargement (left ventricular end-diastolic diameter of 6.4 cm × 8.8 cm) with severe cardiac dysfunction (left ventricular ejection fraction, LVEF of 29.5%). Late gadolinium enhancement (LGE) in the septal ((**A**), white solid arrows) and apex (**B**), white dashed arrows) of LV was revealed. Enlarged mediastinal and bilateral hilar lymph nodes were also noted. Lymphoma or other malignant diseases were suspected. Hence, whole-body^18^F-FDG and ^68^Ga-FAPI-04 PET/CT were performed on two consecutive days for tumor screening.

**Figure 2 diagnostics-13-01450-f002:**
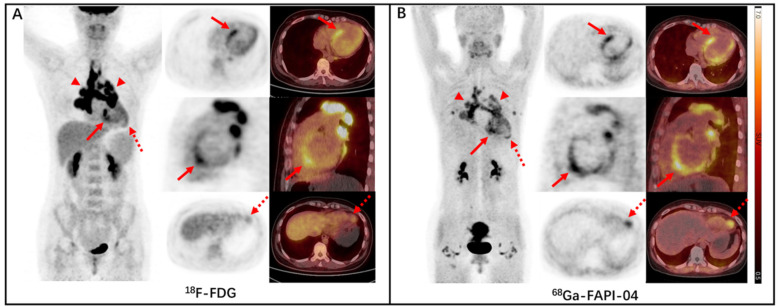
The oncology ^18^F-FDG PET with 4 h fasting time (without the preparation of high-fat low-carb diet) presented mild heterogeneous uptake in the LV wall, particularly elevated in the septum ((**A**), red solid arrows, SUVmax of 5.2). Mild uptake was observed in the apex ((**A**) red dashed arrows, SUVmax of 2.7). ^68^Ga-FAPI-04 PET showed significant diffuse heterogeneous uptake in the LV wall, indicating global cardiac fibroblast activation. The most prominent uptake was detected in the septum ((**B**), red solid arrows, SUVmax of 6.0) and apex ((**B**) red dashed arrows, SUVmax of 4.0), corresponding with the LGE regions shown by CMR. Intense uptake was also detected in mediastinal and bilateral hilar lymph nodes ((**A**), red arrowheads, SUVmax of 14.8 for ^18^F-FDG; (**B**), red arrowheads, SUVmax of 8.0 for ^68^Ga-FAPI-04). An endomyocardial biopsy was conducted. The presence of non-caseating epithelioid cell granulomas on the histological specimens was detected, demonstrating the diagnosis of sarcoidosis. Sarcoidosis is a multisystemic inflammatory granulomatous disease that can manifest with nonspecific clinical pictures. The diagnosis of sarcoidosis could be very challenging and a thorough evaluation [1], through the extensive overview of the disease, has been reported. The clinical course of sarcoidosis varies from a benign to life-threatening disease. An increased risk of pulmonary embolism was associated with sarcoidosis in the presence of antiphospholipid antibodies in these patients [2]. Additionally, cardiac involvement could present a life-threatening situation such as conduction disturbance, ventricular tachyarrhythmia, and congestive heart failure, while the clinical manifestations may be concealed [3]. ^68^Ga-FAPI PET has emerged as a promising imaging tool in various tumor entities and benign diseases. Studies have demonstrated varying degrees of ^68^Ga-FAPI uptake in liver fibrosis, inflammatory bowel disease, arthritis, IgG4-related disease, and other benign conditions [4]. ^68^Ga-FAPI signal was also associated with myocardial ischemia injury upon acute myocardial infarction [5,6,7], dilated cardiomyopathy [8], hypertensive heart disease [9], immune checkpoint inhibitors-associated myocarditis [10], or systemic sclerosis [11]. According to the literature, ^68^Ga-FAPI-04 tracer uptake was increased in systemic sclerosis-related myocardial fibrosis and myocardial biopsies from cardiac MRI-negative and ^68^Ga-FAPI-04-positive regions confirmed the accumulation of FAP+ fibroblasts surrounded by collagen deposits, suggesting that ^68^Ga-FAPI PET may be a diagnostic option to monitor cardiac fibroblast activity. In addition, a recent case reported that pronounced chronic activity of cardiac fibroblast activation was detected by ^68^Ga-FAPI-46 PET/MR in a patient with cardiac sarcoidosis, who was treated with immunomodulatory therapy after 6 months [12]. At the initial diagnosis of cardiac sarcoidosis, ^18^F-FDG PET/MR showed intense FDG uptake and LGE in the septal and posterior wall of the left ventricle in this patient. After 6 months of treatment with corticosteroids and heart failure medication, the follow-up ^18^F-FDG PET revealed no tracer uptake. However, ^68^Ga-FAPI-46 PET/MR demonstrated pronounced tracer uptake and LGE in the basal septum/posterior wall, which matched the regions of ^18^F-FDG/LGE uptake from 6 months before, suggesting ongoing cardiac remodeling. Our current findings demonstrate that ^68^Ga-FAPI-04 PET might have potential utility for sarcoidosis evaluation, particularly in the involvement of the myocardium, which could be masked by physiological ^18^F-FDG uptake.

## Data Availability

No new data were created or analyzed in this study. Data sharing is not applicable to this article.

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
