# Peer review of "In Vivo Fibroblast Activation of Systemic Sarcoidosis: A 68Ga-FAPI-04 PET/CT Imaging Study"

_diagnostics, 2023, doi:10.3390/diagnostics13081450_

Round 1

Reviewer 1 Report

well written paper,

i have only one question:

In the MIP image of FAPI there are some bilateral inguinal lymph nodes.

but they are not described in the article. Why?

Author Response

Response: Thanks. Actually, that were the uptake of the soft tissue around the articulatio coxae, so we didn’t describe it in the article. Please see the attachment file.

Reviewer 2 Report

The "interesting image" submission by Wang et al is concerned with the recent topic of PET imaging using 68Ga-FAPI-04 in nuclear medicine. The authors provide an example of a patient with cardiac dysfunction, who underwent PET/CT with 18F-FDG and 68Ga-FAPI-04, such that a direct comparison could be achieved. The application options of FAP-PET are currently being intensively researched in nuclear medicine, so the authors' example is very timely. In the present case, the patient was diagnosed with sarcoidosis, so that the authors were able to interpret the result of the PET very well and this is certainly highly interesting for the readers of the Diagnistcs.

Some suggestions for improvement of the presentation of the manuscript are: The authors should consistently use the notation "68Ga-FAPI-04" throughout the text and not abbreviate it to "FAPI" to clearly document the difference to other FAPI derivatives. In the context of "myocardial fibrosis in patients with systemic sclerosis," Treutlein et al. recently published a paper (doi: 10.1007/s00259-022-06081-4) that could be briefly discussed, due to the association of sarcoidosis with other autoimmune diseases, such as systemic sclerosis. Similarly, the discussion of Ref. 8 fell somewhat short and could be in some more detail. In some places, the word "The" is missing from the beginning of the sentence. 

Overall, the "interesting image" of Wang et al. is a timely contribution and will certainly arouse the interest of the readership of Diagnostics. 

Author Response

Response: Thank you for your suggestion.

1) We have changed FAPI to 68Ga-FAPI (including 68Ga-FAPI-04 and other derivatives) or 68Ga-FAPI-04 in the corresponding text.

2) We have revised the discussion and added this article (Line 55-59).

3) We have discussed this case in more detail (Line 62-67).

4) We have checked the word “The” in the text.

Reviewer 3 Report

1) L46-48FAPI signal was also associated wih myocardial ischemia injury upon acute myocardial infarc-  tion[2-4], dilated cardiomyopathy[5], hypertensive heart disease[6] or immune checkpoint inhibi- tors-associated myocarditis[7].

a- Sarcoidosis: An Old but Always Challenging Disease. Diagnostics 202111, 696. https://doi.org/10.3390/diagnostics11040696

b- Correlation between Potential Risk Factors and Pulmonary Embolism in Sarcoidosis Patients Timely Treated. J Clin Med. 2021;10(11):2462. Published 2021 Jun 2. doi:10.3390/jcm10112462

2) Please improve the quality of all images.

Author Response

Response:

1) Thank you for your suggestion. We have revised the background and added these references in the text (Line 43-50).

2) Thank you. We have changed the images and uploaded in the corresponding image file.